# Lamb Fattening Under Intensive Pasture-Based Systems: A Review

**DOI:** 10.3390/ani10030382

**Published:** 2020-02-27

**Authors:** Gonzalo Fernandez-Turren, José L. Repetto, José M. Arroyo, Analía Pérez-Ruchel, Cecilia Cajarville

**Affiliations:** 1Departamento de Producción Animal, IPAV, Facultad de Veterinaria, Universidad de la República, Ruta 1 km 42,5 km, San José 80100, Uruguay; joselorepetto@gmail.com (J.L.R.); chemaarroyo@gmail.com (J.M.A.); anapevet@gmail.com (A.P.-R.); ccajarville@fvet.edu.uy (C.C.); 2Programa Nacional de Investigación en Pasturas y Forrajes, Instituto Nacional de Investigación Agropecuaria, La Estanzuela 70002, Colonia, Uruguay

**Keywords:** sheep, fresh forage, intake, rumen, digestion, performance, meat quality

## Abstract

**Simple Summary:**

The use of fresh high-quality pastures in intensive lamb production systems is considered worldwide as economically advantageous, environmentally friendly, and a promoter of animal welfare. Moreover, it generates a desirable meat composition. However, it is known that the availability of pastures in grazing regions is variable throughout the year, and this makes the maintenance of a stable offer of feeds and production difficult. The combination of high-quality pastures and other feedstuffs is very common in dairy cow grazing systems while, for sheep, there is less information available. The objective of the current review is to discuss this topic in light of published information about intake, digestion, rumen environment and health, performance, and carcass quality and composition.

**Abstract:**

The benefits of pasture-based systems on the fatty acid composition of sheep meat appear to be achievable despite variability in the quality of the pastures. Lambs fed high levels of temperate pastures have an excess of N-ammonia derived from protein degradation. Furthermore, animal performance is highly variable depending on the quality of the pasture at the time of grazing, and high animal performance in these systems appears to be linked to the use of high-quality pastures with high availability, and is possibly added to by the inclusion of concentrates that allow increasing energy intake and a better use of the N in the pasture. The combination of high-quality pastures and total mixed ration offers a good alternative to the inclusion of concentrates in the diet, improving the use of N, and avoiding acidosis problems. However, information to determine the effect of a number of nutritional strategies on meat quality, and the minimum level of pasture intake necessary to achieve the benefits of pastoral systems is still lacking.

## 1. Introduction

The meat produced on pasture-based systems is recognized as a high-quality product, with positive nutritional contributions to human health [1,2,3]. In addition, systems that allow animals to graze have a better social perception, as they are associated with natural attributes, environmental care, and animal welfare, which also offer new opportunities for pasture-based systems [4,5,6,7]. For fattening lambs, these systems are developed mainly on grasses (Poaceae) and legumes (Fabaceae). They extend across tropical and temperate regions, but the quality of the forage differs between them. Temperate pastures are, in general, considered to be of high quality, due to the higher digestibility and lower fiber content of temperate compared with tropical species [8]. The most commonly-cultivated pastures include grasses (e.g., *Agrostis* spp., *Festuca* spp., *Lolium* spp., and *Dactylis* spp.) and herbaceous legumes (e.g., *Lotus* spp., *Medicago* spp., and *Trifolium* spp.) [9]. This review will be focused on the use of these type of pastures. 

Temperate-cultivated pastures are usually used for intensive grazing or grazing plus supplementation. Compared with sheep, there is abundant information on the use of nutrients for beef cattle and dairy cows in these rearing conditions [10]. Although sheep have been used as an experimental rumen model for feed evaluation and feeding studies, the information on diets specifically designed for sheep is not as abundant as that for other ruminants (dairy and beef cattle), which may be because of the variety of conditions in which sheep are raised. However, the digestive physiology of sheep is quite different from cattle; mainly associated with differences in digesta retention times, rumen digestion, and fermentation characteristics [11,12,13], so the use of feeding strategies designed for beef or dairy cattle is not recommended for sheep nutrition. For example, Aguerre et al. [13] observed in animals consuming lotus (*Lotus corniculatus*), that increasing supplementation with sorghum grain from 1% to 1.5% of body weight (BW) was effective in increasing feed intake and digestive use in cattle. Nevertheless, the same levels of supplementation in sheep resulted in excessive ruminal fermentation that reduced fiber digestibility and intake. In the same way, van Gastelen et al. [11] highlighted the caution needed before extrapolating results of CH_4_ mitigation, mainly due to differences between dairy cattle, beef cattle, and sheep.

Grazing sheep occupy an acreage that may be limited [14,15] in order to increase areas of native fauna and flora protection. It is also imperative to reduce greenhouse gas emission intensity (CH_4_ and N_2_O released per kg of meat or milk produced), and therefore, it is necessary to increase productivity on pastures worldwide. Low dry matter (DM) intake has been identified as a main constraint of pasture-based systems for high-production animals [16]. In turn, the growth of lambs is conditioned to climatic conditions and excellent grass management is required. Combining pastures with other feedstuffs will allow for the overcoming of restrictions of grazing systems, taking advantage of the benefits of both pasture and confinement.

## 2. The Advantages of Pastures on Meat Characteristics

Recent reviews have been published reporting information on the impact of grazing on the quality of lamb meat [17,18]. Fattening lambs to pasture could be considered as an alternative to produce high quality lamb meat [19], but it is not clear what would be the minimum forage inclusion level at which we could observe improvement the quality of the meat, without limiting the growth, with respect to the lambs reared with concentrates. The demand for healthier and environmentally sustainable meat products has stimulated consumers’ interest in more extensive systems [20]. Lower fat content is, in general, characteristic of the meat of animals fattened on pastures, and is one of the points that has raised more interest. In general, lambs fed on concentrates produce meat with higher fat content than those fed forage-based diets at the same slaughter weight [21,22,23]. On the other hand, it is necessary to point out that fatty acid composition plays an important role in the definition of meat quality, as it is related to the nutritional value of fats for human consumption—these fatty acids have a broad range of biological actions involving cell membrane integrity, signal transduction, gene expression, and prevention of CVD, metabolic and inflammatory diseases, and cancer [3,24]—as well as with differences in organoleptic attributes, especially taste [24,25], even with undesired sensory characteristics [23]. 

In particular, forage degradation in the rumen is a complex process that involves multiple microorganisms. In fact, it has been observed that sheep fed only with fresh ryegrass (*Lolium multiflorum*) have greater microbial diversity than those fed with hay and concentrate [26]. The microbiome of pasture-fed ruminants has the ability, through its enzymatic activity, to synthesize the long-chain n-3 fatty acids (FA) (eicosapentaenoic and docosahexaenoic acid) from the α-linolenic acid precursor [27]. In turn, forage-based diets would favor the growth of fibrolytic microorganisms that are primarily responsible for the intensive hydrogenation activity in the rumen and, consequently, for the production of conjugated linoleic acid (CLA) and vaccenic acid (C18: 1 trans 11, precursor of CLA in tissue) that would have benefits for meat quality [28]. 

Forage species of the pasture may also have effects on the composition of intramuscular fat deposited. In this sense, Fraser et al. [29] compared fatty acid composition of meat derived from lambs fed legumes or grasses, and observed higher concentrations of unsaturated fatty acids when lambs were fed legumes, thus increasing the polyunsaturated:saturated ratio. Meanwhile, De Brito et al. [30] evaluated different herbage species and mixtures in fattening lambs and observed that some mixtures; such as chicory (*Chichorium intybus*) + arrowleaf clover (*Trifolium vesiculosum*), had the ability to promote a better fatty acid profile, while others, such as alfalfa (*Medicago sativa*) + phalaris (*Phalaris aquatica*), led to high Vitamin E concentrations, improving oxidative stability, reducing the conversion of oxymyoglobin to metmyoglobin and, as a consequence, improving the shelf life of meat. 

Finally, it is necessary to point out that, frequently in the literature, the effect of the type of diet appears to be confounded by the production system. For example, Popova et al. [31,32], concluded that feeding pastures decreases fat and increases n-3 polyunsaturated content in the carcass. However, these studies compared indoor vs. grazing animals, with all the differences that both systems involve, aside from diets (exercise, comfort, food availability, etc.).

## 3. Nutrient Intake, Digestion and Rumen Environment of Sheep Consuming Temperate Pastures 

In temperate zones, forage species are characterized by higher protein and lower fiber contents compared with tropical ones, and their nutritive value declines less with age, leading to high DM digestibility, which can exceed values of 90% [8]. The chemical composition of temperate pastures and their fermentative characteristics determine the ruminal environment. In turn, it is expected that the inclusion of other feedstuffs in the diets of lambs that consume fresh forage affects the conditions of the rumen environment. Table 1 shows gathered information on the intake, digestibility, and ruminal environment of sheep that consumed diets based on fresh forage. 

The studies presented in Table 1 were performed on ryegrass, oats (*Avena sativa*), and barley (*Hordeum vulgare*) as grasses; white (*Trifolium repens*) and red clover (*Trifolium pratense*), alfalfa, and lotus as legumes; and their mixtures. Chicory was also found in mixtures with grasses and legumes. On average, the neutral detergent fiber (NDF) and acid detergent fiber (ADF) values were (440 ± 94 and 244 ± 35 g/kg DM, respectively; mean ± SD). There was a great variability in water-soluble carbohydrate (WSC) concentration (114 ± 70 g/kg DM; mean ± SD) due to the high variation in ryegrass varieties high in WSC content. The relatively low content of WSC is a characteristic of temperate pastures, which is in general considered a limiting factor to the efficiency of N use at rumen level [44,49,50].

The pastures used (Table 1) had average N values of 28.6 ± 7.1 g/kg DM (mean ± SD), and although no data on protein fractionation are reported in these studies, high values of the soluble protein would be expected [51,52], which, in addition to the WSC content, can explain the high N-ammonia concentration in the rumen, as shown in most experimental treatments (20.0 mg/dL ± 8.8; mean ± SD). High selective grazing of sheep may even increase N-ammonia in rumen, as observed by Pérez-Ruchel et al. [53] comparing lambs grazing with lambs fed the same pasture cut and provided in feeders. Rumen fermentation of temperate pastures is usually high [54,55], indicating a good quality of fiber, which explains the high volatile fatty acid (VFA) concentration in the rumen observed in most studies. In sheep consuming temperate pastures as sole feed, the average VFA concentration varied from 58 to 165 Mm, with a net predominance of acetate in most of the studies (Table 1). Only a few papers report microbial protein synthesis in the rumen of lambs or sheep on pastures, and low values seem to be related with restricted feeding regimes, more than to pasture or diet composition.

Some authors proposed that pastures composed of mixtures of grasses and legumes can be a strategy to increase intake, preserving forage quality with maturity in comparison to grass monocultures [56]. Niderkorn et al. [35] studied the supply of different proportions of ryegrass and white clover in cannulated sheep. These authors concluded that a mixture containing 25% to 50% of white clover led to better results with respect to only ryegrass forage. Lambs fed only white clover consumed more, but had high concentrations of N-ammonia, leading to greater N losses. According to Rutter [57], sheep and cattle develop preferences for clover over grass mainly during the morning, but this is reversed during the afternoon. Although the mechanisms involved in these preferences are not clear enough, release of ammonia from the soluble protein fraction of the forage, and subsequent uptake in the blood, in addition to propionate, have been proposed as inductors of satiety in grazing ruminants [58]. Therefore, combining grasses and legumes in the sward could help to maintain intake during a longer period. On the other hand, the inclusion of legumes containing condensed tannins could reduce the rumen degradation of the protein and increase the flow of protein into the duodenum, which in turn can enhance the efficiency of protein digestion by ruminants [55,59,60,61].

## 4. Performance of Lambs on Pasture-Based Systems

For finishing lambs at pasture, growth rates of 250–300 g/d from birth to weaning and 150–200 g/d from weaning to finishing could be set as a benchmark according the study published for Orr et al. [62]. However, the performance and nutrient intake from temperate pasture is highly variable. For example, some authors report daily gain of 141 g/d [63], while others report 243 g/d [29], both with lambs grazing on alfalfa. We will focus on factors that allow optimizing the intake of fresh forage in lambs.

In Table 2, growth performance results of lambs grazing on different species of temperate pastures, and pastures supplemented with other feedstuffs, are shown.

The data presented in Table 2 show a high variability in the growth performance of lambs fed only fresh forage, ranging from 47.1 to 336 g/d. In this sense, forage-species grazing was found to significantly affect the liveweight gain of the lambs [34]. This great diversity in productive results, in general, is explained by variations in intake [77], although it is necessary to point out that the procedures to measure intake of pasture differed among studies. It is known that in herbivores, the control of intake is multifactorial [78]. In grazing animals, the total daily herbage intake depends on the biting rate, bite weight, and the time spent grazing [79,80]. The biting rate and bite weight decrease as forage advances in the state of maturity because the animals spend more time selecting the food [81].

Results show that lambs grazing on a lucerne-dominant perennial pasture performed better than did lambs grazing on annual pasture with supplements during the finishing period [82]. In addition, Fraser and Rowarth [67] underlined the importance of feed quality on animal performance when they evaluated ryegrass, white clover, chicory, plantain (*Plantago lanceolata*), or lotus. These variations in grass quality are mainly explained by the differences in the fiber content and protein of the pastures used. In this sense, the intake of NDF is considered the first limiting factor of the intake in forage diets since it is related to the reticulum–rumen distention [83].

Figure 1 presents the relationship between dry matter intake and average daily gain (ADG) in treatments on lambs fed only fresh forage [33,42,63,64,67,70,73,74].

Even considering that different methods of intake assessment were used in the studies (see Table 2), and that this fact could add variability to the data set, the consumption of DM in grazing lambs explained almost 40% of the ADG (*P* < 0.001). In this sense, the quality of the pasture would play a determining role; observed values above the predicting line, corresponded mostly to legumes with a high content of N (g/kg DM) or varieties of ryegrass with high WSC content.

In addition, when the plants advance in the state of maturity, the retention time of the forage in the rumen is increased, limiting voluntary intake, mainly due to the increase of cell walls and lignified tissues, and the decrease of the protein content [84]. Reduction in the concentration of the cell wall of the forage and/or the increase of its digestibility would allow the improvement of the animal’s performance [85] through, above all, an increase in consumption. Moreover, it is also related to other characteristics such as the structure of the plant, its morphology, its density and height, and its allowance or spatial distribution. The differences between grasses and legumes have been reported in the literature through an extensive review by Luscher et al. [56]. The smaller content of structural components of the cell wall of herbaceous legumes with respect to the grass represents one of the main differences, which is reflected in a greater digestibility of the organic matter, and greater concentrations of net energy and metabolizable protein. Moreover, other authors have reported that forage legumes, such as red clover and alfalfa, offered either fresh or as silage, could increase growth rates in ruminants due to the increased intake of DM, with regard to grasses such as ryegrass [34,86].

In general, voluntary intake of legume forage is 10% to 15% greater than grasses of similar digestibility [56]; this is probably because, in an equivalent phenological state, legumes have a lower resistance to chewing, a faster rate of digestion, and a faster rate of rumen particles, which in turn reduces rumen filling [87]. The inclusion of legumes (red clover and alfalfa) has the potential to improve the quality of the diet of sheep [88], increasing daily gain [29,89], and some legumes, such as lotus, even reduced the total nematode parasite intensities compared to lambs grazing on ryegrass/white clover swards [42,86,90]. Fraser et al. [29] evaluated the performance of lambs fed with clover, alfalfa, or ryegrass. These authors reported daily gains of 243, 305, and 184 g/d in lambs fed alfalfa, red clover, and perennial ryegrass, respectively.

On the other hand, some authors observed that plant species diversity increases feeding motivation [91], such that the mixture of pastures with several species could be considered as positively affecting forage intake and animal performance [68,76]. Niderkorn et al. [35], in the central region of the Alpes (France), studied different mixtures of perennial ryegrass (cv. AberVon) and white clover (cv. Merwi), at different ratios (0, 0.25, 0.50, 0.75, and 1). They observed that the clover/ryegrass ratios of 0.25 and 0.50 optimized consumption and digestion. Moreover, Papadopoulos et al. [88] evaluated the addition of white clover to orchandgrass (*Dactylis glomerata*) pasture on the performance of grazing lambs. These authors reported lower concentrations of ADF and NDF in a mixture (white clover + orchandgrass) vs. orchandgrass, and body weight gain for lambs grazing on the mixture was 40% greater than for lambs grazing on orchandgrass.

Another constraining factor of intake in grazing is when the moisture concentration of the pasture is too high (above 80%) [92]. Some authors reported that high-moisture forages, compared to dry forages (hay), caused a lower passage rate and lower intake in sheep and had a lower digestibility [93]. However, low moisture can also negatively affect intake of fresh pastures. Kenney et al. [94] evaluated the rate of intake in sheep fed kikuyu grass (*Pennisetum clandestinum*); the intake rate of fresh forage decreased as DM content increased, but above 40% DM content the intake rate remained relatively constant.

Grazing management is a key point in intensive pasture-based systems, in this sense, the time of access to forage is very important. The effects of this management on pasture intake have been studied in dairy cattle and beef cattle [79,95], while there is less information for sheep. Pérez-Ruchel et al. [39] studied the effect of a time restriction on access to feed in sheep fed only pasture, and reported an increase in the intake rate when the time of access to forage was reduced from 24 to 6 h/d. However, this increase in the intake rate did not compensate for the lower time of access to forage, leading to a decrease in the total intake. In addition, Iason et al. [96] evaluated the effects of food availability (5.5 and 3 cm sward height) on the ability of grazing sheep to compensate for the restriction of daily grazing time. In response to restricted time, the sheep had a higher rate of intake, achieved mainly via larger bites. The behavioral responses to restricted time allowed the sheep to counteract the reduced daily grazing time only for the tallest sward, but in short swards, the time restriction led to a reduction in total daily intake. In another experiment, Luciano et al. [97] compared performance of lambs fed concentrates in stall, grass at pasture for 8 h, or grass at pasture for 4 h in the afternoon. These authors concluded that growing lambs can tolerate a restriction of grazing duration without detrimental effects on performances. In general, the effect of restriction time on sheep grazing consumption would appear to be determined by increasing the bite rate, as long as it has enough availability, height, and is a type of pasture that allows compensation [98].

In grazing, the preference of lambs for some species of pastures may be affected by changes in chemical composition with the season of the year [99]. In addition, the management of the time of day when the animals are in the pasture can act for or against the ability of the animal to optimize the harvest. Several authors studying this topic [100,101,102,103] observed a higher concentration of non-structural carbohydrates (NSC) when forage was harvested in the afternoon. Probably, the greater concentration on NSC increased the palatability of the pasture, and as a consequence, increased daily gain [74]. In addition, NSC may be an alternative to increase the efficiency of N use, as will be seen later. Ciavarella et al. [104] observed that differences in the WSC concentration of ryegrasses had a significant influence on dietary choices of sheep grazing. In a review, Edwards et al. [105] studied opportunities to improve the diet quality, intake, and performance of animals through manipulation of the partial preference commonly shown by grazing animals for different pasture components. This work highlights the preference patterns and the complexity of plant–animal interactions.

Based on the results discussed in this section, to maximize the intake of fresh forage in the fattening of lambs, an important point is to consider a better efficiency in the forage harvest, associated with the grazing duration and moment of the day in which the pasture is supplied. In turn, to maximize the intake of fresh forage, there must be high availability, and mixtures of legumes and grass seem to be the best options.

## 5. The Addition of Other Feedstuffs to Temperate Forage-Based Diets

### 5.1. Digestion of Temperate Forages Plus Other Feedstuffs

How the forage is combined with other feedstuffs (concentrate or total mixed ration, TMR) can have a large influence on N metabolism and nutrient utilization and improve efficiency of microbial protein synthesis. There is consensus among authors that the synchrony of carbohydrates (CHO) and nitrogen components is key to increase the synthesis of microbial protein; and that the availability of CHO is decisive in this regard, since in rumen there is no endogenous source for energy to compensate temporary imbalances at the rumen level, as for N through recycling [106]. Given the high level of soluble protein components of the pasture—and the fact that it contains relatively low levels of CHOs with fast fermentation rates (WSC), and high levels with slow ones (cell walls)—one way to achieve synchronization would be the addition to the diet of carbohydrates with rapid fermentation through supplementation with grains or by-products [107]. Theoretically, starch would accompany the rapid release of N, and thus would allow the achievement of higher microbial growth with a positive impact on growth.

Nutrient synchronization has been studied in cattle [108,109] and also in concentrate diets for sheep [110,111,112,113]. In grazing lambs, the results have not been consistent. For example, Trevaskis et al. [114] evaluated the effect of nutrient synchronization in a series of experiments—either by the feeding of carbohydrate-based supplements (sucrose and fine-rolled barley grain) to tropical and temperate pasture (kikuyu; *Pennisetum clandestinum*, or ryegrass), or by providing pasture with a higher ratio of carbohydrate/N (kikuyu or ryegrass cut early morning or late afternoon)—on rumen pH, ammonia, and microbial protein synthesis. The results of these studies supported the hypothesis that there are benefits on microbial protein synthesis in synchronizing the availability of rumen-fermentable carbohydrates with N in the rumen, but this is not always associated with significant changes in rumen pH and NH_3_-N concentrations. Meanwhile, Tebot et al. [43] evaluated three forage diets with non-fibrous carbohydrate supplementation (100% fresh temperate forage, 70% forage + 30 barley grain, and 70% forage + 15% barley + 15% molasses-based product). The results indicated that supplementation with starch (barley) or sugars (molasses) in sheep grazing did not improve ruminal N-ammonia captured for microbial protein synthesis. In addition, Aguerre et al. [13] evaluated lotus as fresh forage with different levels of sorghum grain supplementation (0, 5, 10, and 15 g/kg BW) in lambs, and also reported that grain did not improve the use of N-ammonia; moreover, microbial protein synthesis decreased as grain supplementation levels increased, probably because there was a net predominance of starch (sorghum grain) in the supplement, and it was supplied separately to forage. Overall, these results could indicate mismatches in the use of CHO and N due to feeding management of pastures (animals are not consuming the pasture and the supplement at the same time) rather than the quantity of nutrients (CHO and N) ingested throughout the day. Amaral et al. [36] studied the level of starch and the type of protein (high or low degradability) to supplement lambs consuming ryegrass and concluded that the supplement should contain both starch and true protein sources.

Supplementation with cereal grains on pastures of good quality, not only would not appear to be effective in increasing the capture of N, as mentioned, but would also represent an additional disadvantage, linked to the degradation of CHOs and the reduction of ruminal pH as a consequence. This effect is aggravated in sheep, especially due to a greater susceptibility to episodes of ruminal acidosis with respect to cattle [13]. Ruminal pH reduction would affect, among other things, protein degradation, due to a decrease in the activity of proteolytic enzymes when the pH reaches values below 5.5 [115].

Pasture supplementation with TMR (mixture of forage and concentrates) appears as a strategy that can improve the uptake of N-ammonia and improve microbial protein synthesis on a forage-based diet in cattle [116]; but in sheep, this approach has been little explored. Our group evaluated different levels of fresh forage supplementation to a total mixed ration (TMR) (0, 0.50, 0.75, and only fresh forage) in lambs. We observed that the ruminal pH and ammonia concentration increased as the inclusion of forage in the diet [37]. On the other hand, when cereal grains were replaced by corn by-products combined with fresh forage provided for 8 h per day, rumen pH and ammonia decreased, but the microbial protein synthesis did not change [38]. Figure 2 shows the relationship observed in several studies of our group between N intake and rumen NH_3_-N in lambs fed forage, forage plus concentrate, or TMR. Ruminal NH_3_-N concentration does not seem to be related up to a certain level of N intake.

In addition to the degradation kinetics of CHO sources, there are other factors, such as ruminal pH and the amount of concentrate consumed, that influence the animal and therefore the productive results. Based on the results shown, it seems that such synchrony could not be achieved when good quality pastures are supplemented exclusively with carbohydrates. In addition to the implications on the use of nitrogenous materials from pasture from an environmental point of view, the low capture of N-ammonia in the rumen generally implies a lack in the synthesis of microbial protein and, therefore, a poor use of diet at the digestive level.

### 5.2. Performance of Lambs Fed Fresh Forage Plus Other Feedstuffs

In pasture-based systems, animal performance in general is limited with respect to the most intensive production systems (feedlot), mainly due to the lower contribution of forage energy to achieve high daily gains. However, it is possible to improve animal performance through different strategies to maximize the consumption of fresh forage, in order to obtain the benefits of pasture consumption over meat quality. In this section, the objective is to update the information on feeding strategies that can achieve high daily gains with high levels of inclusion of fresh pasture.

In confinement systems, daily gains of 350–370 g/d are reported [117,118] and over 300 g/d, even with 60% levels of a fibrous by-product (distillers dried grains with solubles) in the diet [117]. Jacques et al. [70] working with lambs fed with ad libitum concentrate, 60% hay and 40% concentrate, fresh forage cut and offered twice a day, and grazing lambs, reported daily gains of 449, 347, 267, and 295 g/d, respectively. Although forage-fed lambs had lower daily gains and longer termination periods, the use of pasture termination systems would avoid excessively fat carcasses of lambs slaughtered at 47 kg liveweight. In addition, Karnezos et al. [63] evaluated the supplementation with corn (123 and 247 g/d) in lambs grazing on alfalfa. ADG in lambs without supplementation reached 141 g/d, and 169 g/d was obtained for lambs supplemented with 247 g/d of corn. In turn, Devincenzi et al. [69] evaluated performance in lambs grazing on alfalfa, supplemented with barley, or confined (concentrate + hay), reporting ADG of 294, 299, and 314 g/d respectively.

## 6. Combining Pastures with TMR Diets for Finishing Lambs

A strategy less studied in sheep is to feed the lambs with a partial mixed ration (PMR) consisting of the combination of a TMR ration and fresh forage of good quality. The use of PMR, alternating daily grazing periods with periods of access to a TMR, seeks to add the positive aspects of pasture-based systems and confinement. Additionally, the use of PMR could overcome the absence of positive results discussed above when grazing grains are supplemented.

Most of the work done with temperate pastures as the only food, and with the aggregate of concentrates or ration completely mixed, have been carried out in dairy cattle [116,119,120,121]. If these effects could be demonstrated in sheep, this strategy could be used to produce high yields of high-quality lamb meat with the inclusion of high levels of fresh forage.

However, the effects of the use of PMR in lambs have not been conclusive so far. Pérez-Ruchel et al. [37], upon supplementing, with fresh alfalfa, lambs fed with decreasing levels of a TMR, observed an increase in nutrient consumption. Fernandez-Turren et al. [38] evaluated a PMR diet composed of fresh forage (alfalfa), cut and offered for 8 h/d, and TMR (either with cereal grains or fibrous by-products) achieved higher levels of consumption with respect to the use of fresh forage only offered ad libitum throughout the day.

Britos et al. [122], studying PMR diets for lambs composed of TMR and fresh alfalfa in a rumen simulation technique (RUSITEC), concluded that these diets could enhance digestibility and rumen conditions with respect to both TMR and forage, and also that the inclusion of a fibrous energy source in the TMR could enhance microbial synthesis. These results, obtained in vitro, coincide with those obtained in vivo in lambs fed PMR diets [38]. This better synchronization in the supply of nutrients at the ruminal level in lambs fed PMR could explain the high daily gains (300 g/d) when Urioste et al. [72] evaluated the performance with the same diets.

## 7. Conclusions

As a result of investigating different strategies for fattening lambs under pasture-based systems, this study found that the quality of the pasture plays a fundamental role, especially in allowing greater efficiency in the times of fattening. However, several points are still unclear, such as the level of pasture necessary to achieve the benefits of pasture-feeding on meat characteristics, the level and type of pasture to maximize intake, and the combination to reach both attributes at once. Work is still needed to deepen strategies that maximize ruminal efficiency and provide data on the digestive use of diets with a high level of forage inclusion in fattening lambs.

The highly productive systems that seek to incorporate high levels of fresh forage would appear to be linked to the use of pastures of high quality and high availability, and possibly added to by the inclusion of other feedstuffs to increase energy consumption and a better use of the N components of the pasture.

## Figures and Tables

**Figure 1 animals-10-00382-f001:**
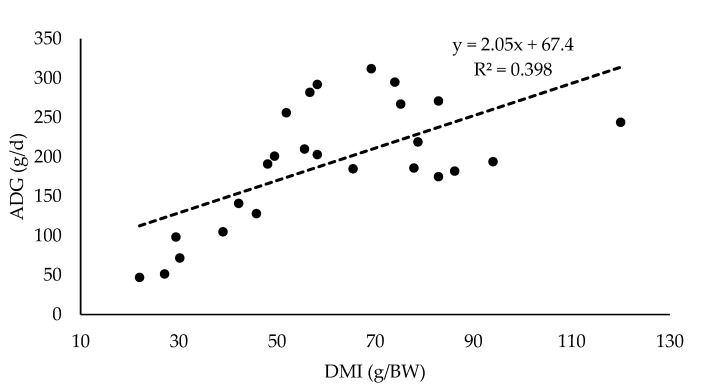
Relationship between dry matter intake (DMI; g/kg BW) and average daily gain (ADG; g/day) in lambs fed only fresh forage.

**Figure 2 animals-10-00382-f002:**
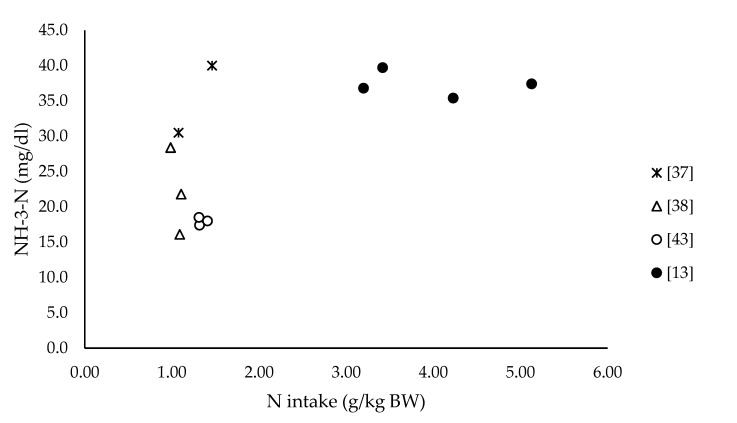
Relationship between N intake and NH_3_-N in lambs fed fresh forage plus feedstuffs (studies plotted measured the intake by difference offered-refused in feeders).

**Table 1 animals-10-00382-t001:** Chemical composition of pasture (g/kg), nutrient intake (g/kg body weight), coefficient of digestibility of dry matter (DMd), rumen environment, microbial nitrogen synthesis (MNS, g/d), and efficiency of microbial nitrogen synthesis (ENMS) in lambs fed fresh forage (FF) or FF plus feedstuffs (concentrate or total mixed ration, TMR).

Diet	FF ^10^	C ^11^	Chemical Composition of Fresh Forage					Nutrient Intake			DMI _m_ ^17^	DMd	Rumen Environment				MNS	ENMS ^21^	Ref
			DM ^12^	N ^13^	NDF ^14^	ADF ^15^	WSC ^16^	DM	N	NDF			pH	NH_3_-N ^18^	VFA ^19^	A:P ^20^			
Ryegrass/white clover	100	L	240	24.8	480	-	121	39.0	0.957	18.7	OR	-	6.44	17.0	79.0	4.10	-	-	[33]
White clover	100	L	146	44.6	264	-	145	51.9	2.31	13.7	OR	-	6.35	28.5	96.0	3.30	-	-	[33]
Alfalfa	100	L	212	39.1	323	-	123	48.1	1.88	15.5	OR	-	6.37	27.5	97.0	3.40	-	-	[33]
Lotus	100	L	-	-	-	-	57.0	38.2	-	-	NA	0.749	-	-	-	-	-	-	[34]
Alfalfa	100	L	-	-	-	-	38.0	50.6	-	-	NA	0.763	-	-	-	-	-	-	[34]
Red clover	100	L	-	39.4	-	-	42.0	45.0	-	-	NA	0.745	-	-	-	-	-	-	[34]
Ryegrass	100	L	-	30.2	-	-	112	27.7	-	-	NA	0.732	-	-	-	-	-	-	[34]
Alfalfa	100	L	-	-	-	-	-	54.6	-	-	NA	0.755	-	-	-	-	-	-	[29]
Red clover	100	L	-	-	-	-	-	65.4	-	-	NA	0.783	-	-	-	-	-	-	[29]
Ryegrass	100	L	-	-	-	-	-	36.8	-	-	NA	0.745	-	-	-	-	-	-	[29]
Ryegrass	100	L	211	24.3	482	219	-	32.9	0.801	15.9	SE	0.757	6.01	14.6	85.8	2.96	-	-	[35]
75% Ryegrass + 25% white clover	100	L	193	28.5	458	218	-	33.9	0.966	15.5	SE	0.765	6.04	18.3	98.1	2.93	-	-	[35]
50% Ryegrass + 50% white clover	100	L	175	32.6	434	218	-	35.9	1.17	15.6	SE	0.766	6.07	26.5	104	3.01	-	-	[35]
25% Ryegrass + 75% white clover	100	L	157	36.8	410	217	-	36.3	1.34	14.9	SE	0.771	6.57	26.5	106	3.02	-	-	[35]
White clover	100	L	139	41.0	386	216	-	36.9	1.51	14.2	SE	0.770	6.05	33.3	115	3.10	-	-	[35]
Ryegrass	100	L	168	34.1	503	240	-	29.3	1.03	14.7	OR	0.710	-	-	-	-	12.1	24.0	[36]
Ryegrass + CM ^1^	79.1	L	168	34.1	503	240	-	31.3	0.923	-	OR	0.750	-	-	-	-	13.4	23.0	[36]
Ryegrass + CM + CG ^2^	80.5	L	168	34.1	503	240	-	33.5	1.19	-	OR	0.720	-	-	-	-	13.3	22.0	[36]
Ryegrass + CM + CC ^3^	80.3	L	168	34.1	503	240	-	33.2	1.19	-	OR	0.750	-	-	-	-	13.2	22.0	[36]
Ryegrass + CG ^4^	81.9	L	168	34.1	503	240	-	35.2	1.27	-	OR	0.730	-	-	-	-	12.9	21.0	[36]
Alfalfa	100	L	296	33.4	374	211	96.8	42.8	1.46	14.9	OR	0.690	6.04	17.6	165	1.41	-	-	[37]
Alfalfa + TMR75 ^5^	62.0	L	296	33.4	374	211	96.8	35.2	1.08	11.9	OR	0.730	6.27	23.5	161	1.60	-	-	[37]
Alfalfa + TMR50 ^6^	44.0	L	296	33.4	374	211	96.8	36.4	1.19	11.8	OR	0.700	6.20	30.5	179	1.58	-	-	[37]
Alfalfa	100	L	214	36.2	366	223	-	27.2	0.985	8.97	OR	0.688	6.49	28.4	146	3.19	-	-	[38]
Alfalfa + TMRa ^7^	46.0	L	214	36.2	366	223	-	38.0	1.11	11.65	OR	0.753	6.31	21.8	154	2.46	-	-	[38]
Alfalfa + TMRf ^8^	46.7	L	214	36.2	366	223	-	37.9	1.10	13.5	OR	0.690	6.14	16.1	152	2.01	-	-	[38]
Lotus	100	W	318	20.2	418	288	-	40.9	0.821	17.1	OR	0.680	6.80	37.4	90.4	3.50	17.6	14.1	[13]
Lotus + 5 g/kg sorghum grain	87.6	W	318	20.2	418	288	-	35.6	0.635	13.8	OR	0.678	6.46	35.4	98.5	3.50	12.7	11.6	[13]
Lotus + 10 g/kg sorghum grain	70.3	W	318	20.2	418	288	-	32.3	0.461	11.2	OR	0.706	6.09	39.7	113	3.26	11.9	12.1	[13]
Lotus + 15 g/kg sorghum grain	58.2	W	318	20.2	418	288	-	32.3	0.384	10.2	OR	0.752	6.16	36.8	141	3.21	10.5	9.29	[13]
Lotus + clover + ryegrass	100	W	-	20.5	444	285	-	24.0	0.566	10.7	OR	0.640	6.65	23.7	-	-	7.16	9.52	[39]
Lotus + clover + ryegrass	100 *	W	-	20.5	444	285	-	17.7	0.397	7.84	OR	0.610	6.66	23.5	-	-	3.19	6.54	[39]
Ryegrass diploid (spring)	100	W	144	24.3	467	244	253	24.5	0.594	11.4	OR	-	-	-	-	-	-	-	[40]
Ryegrass high-sugar (spring)	100	W	166	22.4	418	227	304	27.2	0.609	11.4	OR	-	-	-	-	-	-	-	[40]
Ryegrass tetraploid (spring)	100	W	189	25.6	458	246	229	22.7	0.582	10.4	OR	-	-	-	-	-	-	-	[40]
Ryegrass diploid (autumn)	100	W	177	33.1	631	298	133	22.0	0.727	13.9	OR	-	-	-	-	-	-	-	[40]
Ryegrass high-sugar (autumn)	100	W	183	35.7	580	300	135	23.1	0.820	13.4	OR	-	-	-	-	-	-	-	[40]
Ryegrass tetraploid (autumn)	100	W	160	38.1	611	290	128	20.9	0.797	12.8	OR	-	-	-	-	-	-	-	[40]
Italian ryegrass	100	L	243	27.5	502	-	300	64.7	0.739	32.5	OR	0.661	-	-	-	-	-	-	[41]
Barley	100	L	262	25.4	557	-	186	64.9	0.731	36.2	OR	0.783	-	-	-	-	-	-	[41]
Alfalfa	100	L	223	34.1	356	277	53.0	55.6	1.51	19.8	NA	0.700	-	-	-	-	-	-	[42]
Red clover	100	L	150	34.9	263	198	74.0	58.2	1.74	15.3	NA	0.730	-	-	-	-	-	-	[42]
White clover	100	L	134	39.8	223	185	90.0	56.7	2.13	12.6	NA	0.730	-	-	-	-	-	-	[42]
Ryegrass	100	L	187	29.8	363	198	166	49.5	1.30	18.0	NA	0.750	-	-	-	-	-	-	[42]
Oats + white clover	100 *	E	159	18.6	554	296	47.0	15.5	1.33	8.57	F	0.670	-	-	-	-	6.29	16.4	[43]
Oats + white clover + Barley	70.0 *	E	159	18.6	554	296	47.0	15.5	1.32	6.74	F	0.720	-	-	-	-	6.09	13.9	[43]
oats + white clover + barley + MBP ^9^	70.0 *	E	159	18.6	554	296	47.0	15.5	1.42	6.40	F	0.710	-	-	-	-	5.93	13.4	[43]
Oats + white clover	100 *	E	148	23.7	546	279	82.0	15.5	1.85	8.44	F	0.710	6.33	17.4	-	-	8.30	20.8	[43]
Oats + white clover + barley	70.0 *	E	148	23.7	546	279	82.0	15.5	1.52	6.67	F	0.750	6.15	18.5	-	-	8.09	16.7	[43]
oats + white clover + barley + MBP ^9^	70.0 *	E	148	23.7	546	279	82.0	15.5	1.64	6.31	F	0.710	6.51	18.0	-	-	7.55	16.2	[43]
Mixed grass + legumes (forage cut at 7:00 h)	100 *	L	147 *	23.0 *	499 *	268 *	14.4 *	22.1	0.510	6.47	OR	-	6.47	17.8	92.5	2.34	4.71 **	11.6 **	[44]
Mixed grass + legumes (forage cut at 18:00 h)	100 *	L	147 *	23.0 *	499 *	268 *	16.8 *	20.0	0.460	6.29	OR	-	6.28	18.5	95.4	2.36	4.16 **	14.5 **	[44]
Ryegrass	100	W	203	19.8	505	248	-	24.7	0.494	12.4	SE	0.767	-	7.81	92.7	3.20	-	-	[45]
Ryegrass + chicory	100	W	153	21.3	430	228	-	27.4	0.743	11.9	SE	0.766	-	16.0	100	3.20	-	-	[45]
Ryegrass + white clover + chicory	100	W	156	26.8	435	231	-	29.3	0.627	12.4	SE	0.765	-	7.92	99.7	3.40	-	-	[45]
Chicory	100	W	103	22.7	353	208	-	30.9	0.698	10.5	SE	0.773	-	6.86	104	3.60	-	-	[45]
Chicory	100 *	W	119	18.2	239	188	153	13.3	0.256	3.18	OR	0.765	6.44	9.80	95.2	3.21	-	-	[46]
Chicory	100 *	W	119	18.2	239	188	153	22.3	0.407	5.32	OR	0.761	6.30	5.60	106	3.75	-	-	[46]
Ryegrass	100 *	W	165	31.5	423	218	114	13.7	0.426	5.80	OR	0.741	6.35	28.3	83.8	3.67	-	-	[46]
Ryegrass	100 *	W	165	31.5	423	218	114	21.1	0.680	8.91	OR	0.753	6.18	27.1	92.2	3.38	-	-	[46]
Ryegrass	100	L	148	29.0	464	242	83.0	25.0	0.730	11.6	OR	0.646	6.71	-	74.5	3.53	-	-	[47]
Ryegrass	100	L	198	25.6	445	231	123	31.5	0.810	14.0	OR	0.750	6.71	-	58.3	2.93	-	-	[47]

^1^ 7 g/kg LW daily of cassava meal; ^2^ cassava meal plus corn gluten meal; ^3^ cassava meal plus calcium caseinate; ^4^ corn gluten feed; ^5^ TMR at a level of 0.75 of the potential intake; ^6^ TMR at a level 0.50 of the potential intake; ^7^ TMR with cereal grains; ^8^ TMR with by-products; ^9^ molasses-based product; ^10^ level of FF intake of diets; ^11^ category of sheep (L: lambs; W: wethers; E: ewes); ^12^ dry matter; ^13^ nitrogen; ^14^ neutral detergent fiber; ^15^ acid detergent fiber; ^16^ water soluble carbohydrates; ^17^ DMI_m_: methods of intake measure (OR: feed offered less feed refused; NA: n-alkane; SE: sensors; F: Intake was individually fixed); ^18^ mg/dL; ^19^ total volatile fatty acids (mM); ^20^ acetate:propionate ratio; ^21^ expressed relating to the apparently digestible organic matter ingested; * Restricted forage was offered; ** data available in Thesis Pérez-Ruchel [48].

**Table 2 animals-10-00382-t002:** Dry matter intake (DMI, g/kg body weight (BW)) and average daily gain (ADG, g/d) of sheep fed fresh forage (FF) or FF plus feedstuffs (concentrate or total mixed ration, TMR) with different biomass allowance (t DM/ha) and chemical composition (g/kg BW).

Diet	FF ^9^	Allowance	C ^10^	Chemical Composition of Forage					N ^16^	Breed	BW ^17^	DMI	DMI _m_ ^18^	ADG	Ref
				DM ^11^	N ^12^	NDF ^13^	ADF ^14^	WSC ^15^							
Ryegrass var. AberDart (continuous grazing system)	100 *	1.67	L	220	32.3	537	284	115	20	Brecknock Cheviot	25.5	22.0	NA	47.1	[64]
Ryegrass var. AberDart (rotational grazing system)	100	2.13	L	187	33.4	482	263	113	20	Brecknock Cheviot	25.5	29.4	NA	98.4	[64]
Ryegrass var. Fennema (continuous grazing system)	100 *	1.57	L	216	34.1	533	278	100	20	Brecknock Cheviot	25.5	27.1	NA	51.5	[64]
Ryegrass var. Fennema (rotational grazing system)	100	2.15	L	188	30.1	496	271	100	20	Brecknock Cheviot	25.5	30.2	NA	71.7	[64]
Ryegrass high WSC (spring)	100	-	L	-	34.6	434	-	196	40	-	42.0	-	-	170	[65]
Ryegrass diploid (spring)	100	-	L	-	34.1	458	-	182	40	-	42.0	-	-	158	[65]
Ryegrass tetraploid (spring)	100	-	L	-	36.0	440	-	178	40	-	42.0	-	-	164	[65]
Ryegrass high WSC (autumn)	100	-	E	-	40.3	416	-	186	40	-	30.7	-	-	179	[65]
Ryegrass diploid (autumn)	100	-	L	-	42.7	439	-	155	40	-	30.7	-	-	179	[65]
Ryegrass tetraploid (autumn)	100	-	L	-	43.8	416	-	162	40	-	30.7	-	-	206	[65]
Ryegrass high WSC (spring)	100	-	L	-	28.0	462	-	251	60	-	28.7	-	-	133	[65]
Ryegrass diploid (spring)	100	-	L	-	28.3	497	-	221	60	-	28.7	-	-	120	[65]
Ryegrass tetraploid (spring)	100	-	L	-	30.2	485	-	213	60	-	28.7	-	-	118	[65]
Ryegrass	100	1.80	L	-	25.6	450	270	80.0	400	-	-	-	-	215	[66]
High sugar ryegrass	100	1.80	L	-	25.6	450	270	96.0	400	-	-	-	-	238	[66]
Ryegrass	100	1.80	L	-	25.6	450	270	80.0	400	-	-	-	-	171	[66]
High sugar ryegrass	100	1.80	L	-	25.6	450	270	109	400	-	-	-	-	210	[66]
Alfalfa	100	1.12	L	223	34.1	356	277	53.0	24	Suffolk × Mule	27.5	55.6	NA	210	[42]
Red clover	100	1.32	L	150	34.9	263	198	74.0	24	Suffolk × Mule	27.5	58.2	NA	292	[42]
White clover	100	1.37	L	134	39.8	223	185	90.0	24	Suffolk × Mule	27.5	56.7	NA	282	[42]
Ryegrass	100	0.940	L	187	29.8	363	198	166	24	Suffolk × Mule	27.5	49.5	NA	201	[42]
Chicory	100	-	L	-	38.9	-	-	-	20	Coopworth	22.5	86.2	Cr_2_O_3_	182	[67]
White clover	100	-	L	-	44.8	-	-	-	20	Coopworth	22.5	78.7	Cr_2_O_3_	219	[67]
Lotus	100	-	L	-	41.1	-	-	-	20	Coopworth	22.5	-	Cr_2_O_3_	-	[67]
Ryegrass	100	-	L	-	32.2	-	-	-	20	Coopworth	22.5	45.8	Cr_2_O_3_	128	[67]
Lotus	100	1.43	L	-	-	-	-	-	10	Suffolk × Mule	30.3	-	-	278	[34]
Alfalfa	100	2.16	L	-	-	-	-	-	10	Suffolk × Mule	30.8	-	-	200	[34]
Red clover	100	1.78	L	-	39.4	246	-	-	10	Suffolk × Mule	30.5	-	-	228	[34]
Ryegrass	100	1.41	L	-	30.2	189	-	-	10	Suffolk × Mule	30.3	-	-	182	[34]
Alfalfa	100	2.37	L	-	-	-	-	-	20	Suffolk × Mule	31.5	-	-	243	[29]
Red clover	100	2.19	L	-	31.5	-	-	-	20	Suffolk × Mule	31.5	-	-	305	[29]
Ryegrass	100	1.31	L	-	-	-	-	-	20	Suffolk × Mule	31.5	-	-	184	[29]
Ryegrass	100	1.20	L	-	32.6	426	193	-	30	Suffolk, Texel, others	30.9	-	-	183	[68]
Ryegrass + white clover	100	1.20	L	-	31.8	445	192	-	30	Suffolk, Texel, others	32.5	-	-	192	[68]
6 species of forage ^1^	100	1.20	L	-	31.4	405	194	-	30	Suffolk, Texel, others	33.3	-	-	193	[68]
9 species of forage ^2^	100	1.20	L	-	30.2	400	190	-	30	Suffolk, Texel, others	32.1	-	-	193	[68]
Alfalfa	100	-	L	-	39.8	-	-	-	36	Rambouillet × Suffolk	30.7	42.2	Cut	141	[63]
Alfalfa + 123 g corn	-	-	L	-	39.5	-	-	-	36	Rambouillet × Suffolk	30.7	41.6	Cut	154	[63]
Alfalfa + 247 g corn	-	-	L	-	39.4	-	-	-	36	Rambouillet × Suffolk	30.7	43.5	Cut	169	[63]
Alfalfa grazing	100	-	L	-	-	-	-	-	12	Romane	21.5	-	-	299	[69]
Alfalfa + barley	62.1	-	L	-	-	-	-	-	12	Romane	21.5	-	-	294	[69]
Dactylis + alfalfa ^3^	100	1.53	L	197	28.5	476	308	-	10	Dorset	23.6	75.2	NDFi	267	[70]
Dactylis + alfalfa ^4^	100	1.53	L	197	28.5	476	308	-	10	Dorset	23.6	74.0	NDFi	295	[70]
Orchandgrass	100	-	L	192	35.7	567	305	-	8	-	25.9	-	-	147	[71]
Ryegrass	100	-	L	158	36.6	489	286	-	8	-	25.5	-	-	152	[71]
Alfalfa	100	-	L	181	47.7	314	234	-	8	-	25.4	-	-	239	[71]
Orchandgrass	100	-	L	209	37.4	559	307	-	4	Suffolk × St. Croix × Ramb	29.2	-	-	149	[71]
Ryegrass	100	-	L	191	33.4	473	283	-	4	Suffolk × St. Croix × Ramb	27.7	-	-	150	[71]
Alfalfa	100	-	L	196	46.6	293	228	-	4	Suffolk × St. Croix × Ramb	22.9	-	-	175	[71]
Orchandgrass	100	-	L	237	39.0	507	284	-	8	-	21.9	-	-	112	[71]
Ryegrass	100	-	L	213	37.9	443	269	-	8	-	21.4	-	-	85	[71]
Alfalfa	100	-	L	230	42.9	281	197	-	8	-	21.8	-	-	256	[71]
Alfalfa + TMRa ^5^	40	-	L	214	36.2	366	223	-	18	Corriedale × Ile de France	29.5	46.9	Cut	336	[72]
Alfalfa + TMRf ^6^	41	-	L	214	36.2	366	223	-	18	Corriedale × Ile de France	29.5	42.6	Cut	305	[72]
Lotus	100	5.27	L	-	56.0	449	350	-	20	Romney	22.8	58.2	Cr_2_O_3_	203	[73]
Alfalfa	100	5.24	L	-	49.8	423	311	-	20	Romney	22.8	65.5	Cr_2_O_3_	185	[73]
Ryegrass WSC	100	-	L	-	23.8	411	222	143	5	Bluefaced Leicester	14.0	69.2	EC	312	[74]
Ryegrass control WSC	100	-	L	-	27.0	487	255	89.0	5	Bluefaced Leicester	14.0	82.9	EC	271	[74]
Ryegrass WSC	100	-	L	-	32.0	473	250	113	5	Bluefaced Leicester	14.0	120	EC	244	[74]
Ryegrass control WSC	100	-	L	-	26.6	540	279	75.0	5	Bluefaced Leicester	14.0	94.0	EC	194	[74]
Ryegrass WSC	100	-	L	-	29.0	506	267	92.0	5	Bluefaced Leicester	14.0	77.9	EC	186	[74]
Ryegrass control WSC	100	-	L	-	31.0	514	274	84.0	5	Bluefaced Leicester	14.0	82.9	EC	175	[74]
*Trifolium alexandrium* + concentrate	65.4	-	L	-	-	-	-	-	9	Lohi	21.0	37.1	Cut	130	[75]
*Trifolium alexandrium* + concentrate	66.0	-	L	-	-	-	-	-	9	Lohi	21.0	37.6	Cut	160	[75]
*Trifolium alexandrium* + concentrate	66.7	-	L	-	-	-	-	-	9	Lohi	21.0	38.6	Cut	180	[75]
Herb/clover ^7^	100	3.45	L	-	25.3	281	-	-	6	Romney	33.0	-	-	247	[76]
Plantain/pasture ^8^	100	3.79	L	-	20.6	399	-	-	6	Romney	33.0	-	-	107	[76]
Tetraploid ryegrass and white clover	100	3.79	L	-	31.4	481	-	-	6	Romney	33.0	-	-	119	[76]
Diploid ryegrass, other grass species, and white clover	100	5.55	L	-	22.6	537	-	-	6	Romney	33.0	-	-	119	[76]
Ryegrass/white clover	100		L	240	24.8	480	-	121	18	-	28.5	39.0	OR	105	[33]
White clover	100		L	146	44.6	264	-	145	18	-	28.5	51.9	OR	256	[33]
Alfalfa	100		L	212	39.1	323	-	123	18	-	28.5	48.1	OR	191	[33]

^1^ perennial ryegrass, timothy, white clover, red clover, plantain, and chicory; ^2^ perennial ryegrass, timothy, cocksfoot, white clover, red clover, birdsfoot trefoil, plantain, chicory, and yarrow; ^3^ cut daily ad libitum; ^4^ rotational grazing; ^5^ TMR with cereal grains; ^6^ TMR with by-products; ^7^ chicory, plantain, red clover, and white clover; ^8^ plantain, ryegrass, and white clover; ^9^ level of FF intake of diets; ^10^ category of sheep (L: lambs; W: wethers; E: ewes); ^11^ dry matter; ^12^ nitrogen; ^13^ neutral detergent fiber; ^14^ acid detergent fiber; ^15^ water soluble carbohydrates; ^16^ lambs per treatment; ^17^ body weight (kg); ^18^ DMI_m_: methods of intake measure (NA: n-alkane; Cr_2_O_3_:chromic oxide; Cut: disappearance herbage mass; NDFi: indigestible NDF; EC: exclosure cages; OR: feed offered less feed refused); * restrict offered.

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
