# Peer review of "Lamb Fattening Under Intensive Pasture-Based Systems: A Review"

_animals, 2020, doi:10.3390/ani10030382_

Round 1

Reviewer 1 Report

Line 45: grasses (e.g. Agrostis spp., Festuca spp., Lolium spp. Dactylis spp.) and herbaceous legumes (e.g. Lotus corniculatus, Medicago spp., Trifolium spp.) [9]. All species are mentioned as genus, why Lotus corniculatus is mentioned as genus and specie. I suggest to follow the same pattern because also could be valid for other Lotus as uliginosus or tenuis.

Table 1: The column of diets requires improvement. There are some aspects that are not clear enough. For example Ryegrass diploid S (the meaning of S)

Line 153: red clover in cannulated sheep. These authors concluded that a mixture containing 25% to 50% of white clover led to better results with respect to only ryegrass forage. Lambs fed only red clover. It is confusing the experiment was with red clover ?

Table 2: The column of diets require improvement. There are some aspects that are not clear enough. For example Ryegrass var. AberDart C (the meaning of C)

Line 238-243: The idea is confusing. Decrease in DM concentration = lower intake. The next paragraph increase in DM increase intake. Need more discussion

Author Response

All suggested changes have been incorporated in the manuscript and highlighted in yellow. Thank you very much for the comments.

Point 1: Line 45: grasses (e.g. Agrostis spp., Festuca spp., Lolium spp. Dactylis spp.) and herbaceous legumes (e.g. Lotus corniculatus, Medicago spp., Trifolium spp.) [9]. All species are mentioned as genus, why Lotus corniculatus is mentioned as genus and specie. I suggest to follow the same pattern because also could be valid for other Lotus as uliginosus or tenuis.

Response 1: Changed, thanks (See line 45).

Point 2: Table 1: The column of diets requires improvement. There are some aspects that are not clear enough. For example, Ryegrass diploid S (the meaning of S).

Response 2: Abbreviations were clarified putting them in full text. Revisiting the table, we found confusing the reference to experimental periods of some papers cited. As the table has the chemical composition of the pasture of each period, we eliminated its references in table 1 and also in table 2 (e.g. in Oats + White Clover 1, 1 was eliminated).

Point 3: Line 153: red clover in cannulated sheep. These authors concluded that a mixture containing 25% to 50% of white clover led to better results with respect to only ryegrass forage. Lambs fed only red clover. It is confusing the experiment was with red clover ?

Response 3: Thanks, it was a mistake and it has been changed. The experiment evaluated only white clover and ryegrass (See lines 153-154).

Point 4: Table 2: The column of diets require improvement. There are some aspects that are not clear enough. For example Ryegrass var. AberDart C (the meaning of C).

Response 4: Abbreviations were clarified, and experimental periods were eliminated (see comment above).

Point 5: Line 238-243: The idea is confusing. Decrease in DM concentration = lower intake. The next paragraph increase in DM increase intake. Need more discussion.

Response 5: We agree with the comment. Our idea was to describe that both too low and too high DM concentration of the pasture negatively affect intake. Please see if our writing is clearer in this version (see lines 242-249).

Reviewer 2 Report

Title: Lamb Fattening Under Intensive Pasture-based Systems: a review

Journal: Animals (MDPI)

The present manuscript, which is a review neatly presented and correctly written, deals with the lamb production systems under intensive pasture-based systems. Under the context of climate change issues, compiling this information is very convenient since it would allow to optimize these production systems, which are environmentally friendly, guarantee the welfare of the animals, and allow improving the meat quality of the animals produced. The information contained in the manuscript is updated with very recent references. My only suggestion is provided below:

-It would be necessary to go deeper into the discussion related to different procedures to measure the dry mater intake when animals are under grazing conditions. This is fundamental to guarantee the quality of the measures performed when feed efficiency is approached.

Author Response

All suggested changes have been incorporated in the manuscript and highlighted in yellow. Thank you very much for the comments.

Point 1: The present manuscript, which is a review neatly presented and correctly written, deals with the lamb production systems under intensive pasture-based systems. Under the context of climate change issues, compiling this information is very convenient since it would allow to optimize these production systems, which are environmentally friendly, guarantee the welfare of the animals, and allow improving the meat quality of the animals produced. The information contained in the manuscript is updated with very recent references. My only suggestion is provided below:

-It would be necessary to go deeper into the discussion related to different procedures to measure the dry mater intake when animals are under grazing conditions. This is fundamental to guarantee the quality of the measures performed when feed efficiency is approached.

Response 1: A specific discussion about the advantages and drawbacks of methods of intake measure is out of topic in this review. However, we agree with the comment respect to the importance of the method used in the prediction of intake, mainly in relation to a specific result. Therefore, we included a column indicating the method employed to measure herbage intake in each table. In addition, we included a comment respect to the diversity of methods employed in the studies and its possible interference with the results (see lines 185-186 and 205-206 and note on Figure 2).